

# Psychological distress, work environment quality, and motivation levels among nurses working in Saudi Arabia

Hanan Alharbi[1], Kholoud Alharbi[2], Ghareeb Bahari[2], Yousef Alshamlani[3] and Regie Buenafe Tumala[4]

[1] Maternity and Child Nursing Department, College of Nursing, Princess Nourah bint Abdulrahman University, Riyadh, Saudi Arabia
[2] Nursing Administration and Education Department, College of Nursing, King Saud University, Riyadh, Saudi Arabia
[3] Nursing Affairs Medical City, King Saud University, Riyadh, Saudi Arabia
[4] Medical Surgical Department, College of Nursing, King Saud University, Riyadh, Saudi Arabia

Corresponding author
Ghareeb Bahari, gbahari@ksu.edu.sa

## ABSTRACT

**Background**. A favorable clinical environment and nurse motivation are both essential for achieving high-quality patient outcomes and organizational performance in healthcare systems, which can be highly distressing for nurses. The purpose of this study was thus to determine the associations among and influences of psychological distress, work environment quality, and motivation on one another.

**Methods**. This was a cross-sectional, descriptive-correlational study conducted with a total sample of 204 nurses in two public tertiary hospitals. We used an online survey to collect nurses' responses, which comprised the Kessler Psychological Distress Scale-6, Brief Nurses' Practice Environment Scale, and Multidimensional Work Motivation Scale. We ran the necessary statistical analyses in SPSS version 28.

**Results**. We found that nurses' psychological distress, work environment quality, and motivation were moderate. A significant, positive, and moderate relationship existed between psychological distress and work environment quality, while nurses' educational level was statistically different with their motivation level. In the regression, only the nationality, current position, and work environment variables significantly influenced psychological distress. The hospital type also significantly influenced work environment quality. No variable was found to influence nurses' motivation level, though.

**Conclusions**. We concluded that nurses' work environment quality and motivation are interconnected with the psychological stress they experience at work. Monitoring nurses' work environment quality, motivation levels, and psychological distress is thus vital to ensure better patient care.

## INTRODUCTION

A healthy work environment is essential for nurses to achieve good patient outcomes and organizational performance (*Mabona, Van Rooyen & Ten Ham-Baloyi, 2022*). Nurses' performance in an organization largely depends on personal motivation, which indicates

the quality of patient care they provide (*Asadi, Memarian & Vanaki, 2019*). Hence, nurses must work in a healthy environment and be motivated in order for them to provide high-quality patient care (*Asadi, Memarian & Vanaki, 2019*). Furthermore, nurses help improve the health of their patients, which can be maximized if nurses operate or work in an environment that is empowering, motivating, safe, and satisfying (*Apex Apeh et al., 2020*). However, because nurses work under high demands and workloads, they are at a high risk of developing psychological distress.

Psychological distress has become a highly debated topic due to its growing significance in understanding and addressing mental health issues (*Jung, Lim & Chi, 2020*). It has been found that work-related stress increases occupational hazards and decreases efficiency inside and outside the work environment (*Jung, Lim & Chi, 2020*). It has also been estimated that work-related stress is a major factor underlying high levels of absenteeism among workers (*Davey et al., 2019*). Stress is a feeling produced by uncontrollable or threatening events that can negatively influence the body's hormonal balance, leading to anger, nervousness, frustration, or anxiety (*Yazdanpanah & Mahin, 2004*).

Work-related stress among nurses was first reported in 1960 when Menzies determined the four causes of anxiety: decision-making, patient care, taking responsibility, and change (*Davey et al., 2019*). Since the mid-1980s, work-related stress among nurses has increased due to the frequent use of technology, increasing healthcare costs (*Jeronimus et al., 2021*), and tension or upheavals within their work environment. Nurses' stress can be categorized into four main types, including (a) social stress, (b) financial stress, (c) academic stress, and (d) clinical area stress (*Aljohani et al., 2021*). It has also been found that stress can affect one's mind, body, and behavior in various ways, and individuals may experience stress differently. Continuous stress among nurses can lead to memory problems and lack of concentration, depression, too much or too little sleep, and high rates of burnout (*Membrive-Jiménez et al., 2022*).

Stress has a major effect on nurses' ability to complete tasks. That is, poor decision-making, a lack of concentration, decreased motivation, and anxiety may impair job performance, causing nurses to make uncharacteristic errors (*Bandura, 1986*). Stress can also significantly contribute to absenteeism, decreased work performance, and, eventually, burnout (*Membrive-Jiménez et al., 2022*). A stressful or negative work environment, along with its undesirable consequences, results from failing to manage conflicts in the hospital organization (*Mabona, Van Rooyen & Ten Ham-Baloyi, 2022*). Additionally, the COVID-19 pandemic has had significant effects on nurses' health statuses due to their increased workload and fear of getting infected, which have caused work-related stress that has affected the nursing work environment. This calls for the wellbeing of nurses to become a health priority (*Li et al., 2024*). On the other hand, a positive work environment enhances the wellbeing of nurses (*Chung et al., 2020*). It has been found that providing a supportive work environment requires professional autonomy and communication, leadership, and teamwork since these are effective at mitigating or solving conflict situations in hospital settings (*Hosseini Moghaddam, Mohebbi & Tehranineshat, 2022*).

Several studies have implied that one reason why nurses are highly motivated to and satisfied at work is good or positive working conditions (*Kagan, Hendel & Savitsky, 2021*).

In the Netherlands, a Delphi study found that having intrinsic motivation and a low level of stress are two of the 36 elements comprising a positive work environment in hospital settings (*Maassen et al., 2021*). In addition, a systematic review reported that patients are highly satisfied with the care provided to them by nurses who have positive perceptions of their hospital environment (*Copanitsanou, Fotos & Brokalaki, 2017*). In Saudi Arabia, a study comprising 1,007 nurses reported that these nurses' hospital institutions had good work environments, particularly in terms of appropriate staffing, authentic leadership, effective decision-making, meaningful recognition, skilled communication, and true collaboration (*Aboshaiqah, 2015*). Further, the nursing work environment in Saudi Arabia has recently undergone significant changes, with efforts to improve working conditions, increase the number of nurses, and enhance the quality of healthcare services (*AL-Dossary, 2022*).

Although evidence from previous studies has shown reports of psychological stress or distress at the workplace and the effects of the working environment and motivation of nurses, studies on the relationships among these variables are lacking. Therefore, the purpose of this study was to determine the associations among and influences of psychological distress, work environment quality, and motivation levels on one another. We aimed to answer the following three main questions: (1) What are the psychological distress, work environment quality, and motivation levels among a sample of nurses working at public hospitals? (2) What are the associations among nurses' psychological distress, work environment quality, motivation level, and demographic variables? (3) How does each of the three main variables influence the others while the demographic variables are controlled for? The findings of this study can contribute to better nursing practices and provide new perspectives that can inform the design and implementation of future studies aimed at improving health outcomes.

## MATERIAL AND METHODS

### Study design and setting

A cross-sectional, descriptive-correlational study was conducted with nurses working at four hospitals in Saudi Arabia. With this design, we collected data at a single time point (cross-sectional) and then analyzed them to determine whether there were relationships among the variables (descriptive-correlational) (*Afrasiabifar et al., 2021*). Hospitals were coded as A, B, C, and D for confidentiality. The included hospitals are university hospitals, which are healthcare facilities affiliated with universities, located in the Riyadh region. They are also considered teaching hospitals, where students are trained and work alongside experienced healthcare professionals. One hospital (D) is a dental hospital, while the other three are general hospitals that treat all medical conditions. The bed capacities of the four hospitals vary but are no less than 500 beds per hospital. These hospitals were selected because they were conveniently accessible.

### Sampling

A convenience sampling method was used to contact nurses working at the above hospitals. We aimed to recruit participants from a diverse range of hospital settings to reduce the

potential for sampling bias and enhance the generalizability of our findings. Hence, Saudi and non-Saudi male and female nurses who had been working at one of the above hospitals for at least one year were eligible to participate in our study. Nurses working in all positions were eligible to participate, as they could provide rich knowledge that could contribute to improved productivity in nursing. However, those with limited English proficiency were excluded, as they may not have understood the survey questions sufficiently. Further, nurses who chose not to participate were excluded. Using the G*Power tool (Heinrich-Heine-Universität, Düsseldorf, Germany), with the statistical power level set at 0.8, an effect size of 0.15, a Cronbach's $\alpha$ of 0.05, and 10 explanatory variables (age, gender, nationality, marital status, educational level, years of work experience, department of work, current position, shift type, and hospital name), a minimum required sample of 118 participants was estimated. To address missing data, 20% was added, which resulted in a final minimum sample size of 142 participants.

## Instrumentation

The questionnaire included demographic questions about participants' age, gender, nationality, marital status, educational level, years of work experience, department of work, current position, shift type, and hospital name. It also included questions assessing participants' psychological distress (*Kessler et al., 2003*), work environment quality (*Sansó et al., 2021*), and motivation level (*Gagné et al., 2015*).

Psychological distress was assessed using the Kessler Psychological Distress Scale-6. This tool includes six items that measure distress caused in the preceding 30 days (*e.g.*, How often did you feel nervous?) (*Kessler et al., 2003*). Responses are rated using a five-point Likert scale with possible answers of 1 = *all the time*, 2 = *most of the time*, 3 = *some of the time*, 4 = *a little of the time*, and 5 = *none of the time*, with higher scores indicating low psychological distress. The instrument has been used internationally and tested in several cultures and languages, including Arabic. The Arabic version has acceptable reliability, and the cutoff point was determined to be 16.25 (Cronbach's $\alpha = 0.81$) (*Easton et al., 2017*). The Kessler Psychological Distress Scale-6 is available for free use (https://www.hcp.med.harvard.edu/ncs/k6_scales.php). We also obtained permission to use the validated version of the scale if needed (S. Easton, personal communication, 2019).

The Brief Nurses' Practice Environment Scale was used to measure the work environment quality variable (*Sansó et al., 2021*). The scale comprises five statements that represent the five dimensions of the Practice Environment Scale of Nursing Work Index. The five statements of the tool are rated using a four-point Likert-type scale, with responses ranging from 1 = *completely disagree* to 4 = *completely agree*. The total score is computed by summing the scores for the five statements and ranges from 5 to 20 (*Sansó et al., 2021*). The reliability of the Brief Nurses' Practice Environment Scale has been established, and it has a Cronbach's $\alpha$ of 0.702. The confirmatory factor analysis of the scale has an excellent fit that evidences its internal validity (*Sansó et al., 2021*). Finally, a cutoff score of <12 means that the practice environment is unfavorable, while a score of >15 means that the practice environment is favorable (*Sansó et al., 2021*). The factors tested in this scale have been shown to be effective in determining nurses' experiences with high burnout and low

satisfaction with their work, particularly when they perceive their practice environment to be unfavorable. Our third author (G.B.) obtained permission to use the scale (L. Galiana, personal communication, 2022).

The motivation level of nurses was assessed using the Multidimensional Work Motivation Scale. This scale includes 19 items with six dimensions and the same factor structure across seven languages (*Gagné et al., 2015*). The scale is answered on a seven-point Likert-type scale, with responses of 1 = *not at all*, 2 = *very little*, 3 = *a little*, 4 = *moderately*, 5 = *strongly*, 6 = *very strongly*, and 7 = *completely* (*Gagné et al., 2015*). Scores range from 19 to 133, with higher scores indicating higher motivation levels. The reliability of the tool is below 0.70 only in German but above 0.80 in the other six languages in which it is available, including Chinese, Dutch, English, French, Indonesian, and Norwegian. Our third author (G.B.) obtained permission to use the scale (M. Gagne, personal communication, 2022).

## Data collection and analysis

Data were collected using an anonymous online questionnaire in a standardized and systematic manner to ensure their validity and reliability. Prior to distributing the questionnaire form, we reviewed the scales' items to ensure their clarity and identify any potential issues. We also addressed factors that may have discouraged potential participants from completing the questionnaire, such as its length and the wording of the items. We collectively confirmed that the items were appropriate and easy to understand. Social media, personal references, and word of mouth were all utilized to help distribute the questionnaire form, as the goal was to reach a large sample.

Data were collected between October 2022 and February 2023, then analyzed using SPSS version 28. Frequency distribution analyses were conducted to identify missing data or errors and to describe the study's sample characteristics and variables. Only the "years of work experience" variable had missing data, which we addressed using the mean imputation method. We nevertheless included "years of work experience" in the analysis, as it could provide insights into the levels of expertise and knowledge that nurses obtain over time. Bivariate analyses, including the independent samples $t$-test, Pearson coefficient correlation ($r$), and one-way analysis of variance (ANOVA), were all conducted as appropriate. Further, multiple linear regression analysis was conducted to determine the relationships among nurses' psychological distress, work environment quality, and motivation level while controlling for the demographic variables. Some demographic variables with several categories were combined into a twofold format for the regression analyses. We also assessed the internal consistency of our structured questionnaire using a commonly used statistical measure called Cronbach's alpha.

## Ethical considerations

Ethical approval was obtained from the institutional review boards of both Princess Nourah bint Abdulrahman University (Ref. number: 22-0445) and King Saud University (Ref. number: 19/0737/IRB). All participants gave their informed consent. Due to limited access to obtain in-person signatures, we provided an online consent form that included the statement, "Your completion of the survey indicates your agreement to participate. Thank

you!'' Participants were also informed that their participation was voluntary and that they had the right to withdraw from the study at any time without penalty. The collected data were kept confidential and only accessed by the research team members.

## RESULTS

### Sample characteristics

Table 1 provides the sample characteristics of the study ($N = 204$). Participants' mean age was 35.81 years ($SD \pm 8.14$ years; range: 22–58 years). Most participants were women (86.6%), married (58.8%), non-Saudi (69.6%), held a bachelor's degree in nursing (75.5%), and worked at hospital A (66.7%). The mean number of years of work experience in nursing was approximately 12 ($SD \pm 7.64$ years; range: 1–35 years). Close to half of the participants (46%) worked in the medical, surgical, or critical unit of their hospital. Most of the sample (68.6%) who worked 12-hour shifts were bedside nurses. Levels were moderate for psychological distress ($M = 19.99$, $SD = 5.63$, range: 6–30), work environment quality ($M = 13.70$, $SD = 2.90$, range: 5–20), and motivation ($M = 72.77$, $SD = 19.95$, range: 19–123). The Cronbach's $\alpha$ of our structured scale was determined to be .86.

### Statistical analyses

Necessary bivariate analysis results are presented in Tables 2 and 3. No statistical differences were found ($p > .05$) when conducting the independent samples $t$-test (Table 2). The Pearson coefficient test reported statistically significant relationships between age and both years of work experience ($r = .711$, $p = .001$) and psychological distress ($r = .140$, $p = .046$). A significant, positive, and moderate relationship was also found between psychological distress and work environment quality ($r = .505$, $p = .001$). More details are presented in Table 3.

When conducting the one-way ANOVA, the results varied for psychological distress, work environment quality, motivation level, and some demographic variables, including marital status, educational level, department, current position, shift type, and hospital. That is, no significant differences were reported between these demographic variables and psychological distress. However, the department ($F [3,203] = 3.670$, $p = .013$) and hospital ($F [3,203] = 3.723$, $p = .012$) were both statistically associated with work environment quality. Further, only nurses' educational level ($F [2,203] = 3.779$, $p = .024$) was statistically associated with their motivation level.

Tables 4, 5 and 6 show the results of the regression analyses of psychological distress, work environment quality, and motivation levels while controlling for the demographic variables. The psychological distress ($F [12,203] = 7.927$, $p < .001$, $R^2 = .332$), work environment quality ($F [12,203] = 7.806$, $p < .001$, $R^2 = .329$), and motivation level ($F [12,203] = 1.827$, $p = .046$, $R^2 = .103$) models were all statistically significant. In Table 4, only nurses' nationality ($\beta = .200$, $p = .013$), current position ($\beta = .143$, $p = .037$), and work environment quality ($\beta = .492$, $p < .001$) significantly influenced their psychological distress. Table 5 shows that nurses' hospital ($\beta = -.188$, $p = .007$) and psychological distress ($\beta = .492$, $p < .001$) significantly influenced their work environment quality. As shown in Table 6, no variable was found to influence motivation levels.

**Table 1** Sample characteristics ($N = 204$).

| Characteristics | N | (%) |
|---|---|---|
| Age (Years) $M = 35.81$, SD $= 8.14$, range: 22–58 | | |
| Gender | | |
| Male | 27 | (13.2) |
| Female | 177 | (86.8) |
| Nationality | | |
| Saudi | 62 | (30.4) |
| Non-Saudi | 142 | (69.6) |
| Marital status | | |
| Single | 76 | (37.3) |
| Married | 120 | (58.8) |
| Divorced/Widow(er) | 8 | (3.9) |
| Education level | | |
| Diploma | 31 | (15.2) |
| Bachelor's | 154 | (75.5) |
| Higher education | 19 | (9.3) |
| Years of experience (Years) $M = 11.91$, SD $= 7.64$, range: 1–35 | | |
| Department working in | | |
| Medical department | 20 | (9.8) |
| Surgical department | 38 | (18.6) |
| Critical units | 36 | (17.6) |
| Other departments | 110 | (53.9) |
| Current position | | |
| Bedside nurse | 140 | (68.6) |
| Charge nurse | 25 | (12.3) |
| Head nurse | 19 | (9.3) |
| Nurse educator | 11 | (5.4) |
| Nurse supervisor | 9 | (4.4) |
| Shift type | | |
| 8 h | 67 | (32.8) |
| 12 h | 99 | (48.5) |
| Other | 38 | (18.6) |
| Hospital name | | |
| Hospital A | 136 | (66.7) |
| Hospital B | 39 | (19.1) |
| Hospital C | 25 | (12.3) |
| Hospital D | 4 | (2) |

**Notes.**
M, Mean; SD, Standard Deviation; SR, Saudi Riyal.

**Table 2 Mean differences between psychological distress, working environment, motivation level, and some demographic variables.**

| Variable mean differences (*t*-test) | Binary categories | Psychological distress | | *p*-value | Working environment | | *p*-value | Motivation level | | *p*-value |
|---|---|---|---|---|---|---|---|---|---|---|
| | | *M* | *(SD)* | | *M* | *(SD)* | | *M* | *(SD)* | |
| Gender | Male | 20.48 | 6.09 | 0.445 | 14.33 | 2.57 | 0.391 | 69.51 | 21.34 | 0.714 |
| | Female | 19.92 | 5.58 | | 13.60 | 2.94 | | 73.27 | 19.75 | |
| Nationality | Saudi | 17.50 | 4.95 | 0.118 | 12.98 | 3.04 | 0.462 | 67.03 | 19.59 | 0.727 |
| | Non-Saudi | 21.08 | 5.58 | | 14.01 | 2.79 | | 75.28 | 19.65 | |

**Table 3 Correlations between continuous variables.**

| | Age | Years of experience | Psychological distress | Working environment | Motivation level |
|---|---|---|---|---|---|
| Age | 1 | | | | |
| Years of experience | **0.711**[*] | 1 | | | |
| Psychological distress | **0.140**[*] | 0.129 | 1 | | |
| Working environment | 0.049 | 0.097 | **0.505**[*] | 1 | |
| Motivation level | 0.078 | 0.129 | 0.128 | 0.130 | 1 |

Notes.
[*]*p*-value $< 0.05$.

**Table 4 Multiple regression analysis for variables associated with psychological distress.**

| Variables | Unstandardized coefficients | | Standardized coefficients | Sig. | Model summary | |
|---|---|---|---|---|---|---|
| | B | Std. error | Beta | | $R^2$ | *P* value |
| (Constant) | 0.032 | 4.458 | | 0.994 | 0.335 | **0.000**[*] |
| Age | 0.061 | 0.068 | 0.088 | 0.372 | | |
| Gender | 0.311 | 1.011 | 0.019 | 0.758 | | |
| Nationality | 2.440 | 0.978 | 0.200 | **0.013**[*] | | |
| Marital Status | −0.528 | 0.752 | −0.046 | 0.483 | | |
| Education level | 0.538 | 0.829 | 0.041 | 0.517 | | |
| Work department | 0.093 | 0.764 | 0.008 | 0.903 | | |
| Years of experience | −0.013 | 0.073 | −0.016 | 0.860 | | |
| Current position | −1.730 | 0.825 | 0.143 | **0.037**[*] | | |
| Shift type | −0.399 | 0.852 | −0.035 | 0.640 | | |
| Hospital | 1.510 | 0.823 | 0.127 | 0.068 | | |
| Work environment | 0.956 | 0.122 | 0.492 | **<0.001**[*] | | |
| Motivation level | 0.014 | 0.018 | 0.049 | 0.430 | | |

Notes.
[*]*p*-value $< 0.05$.

**Table 5  Multiple regression analysis for variables associated with work environment.**

| Variables | Unstandardized coefficients | | Standardized coefficients | Sig. | Model summary | |
|---|---|---|---|---|---|---|
| | B | Std. error | Beta | | $R^2$ | P value |
| (Constant) | 12.511 | 2.117 | | <0.001 | 0.329 | **0.000**[*] |
| Age | −0.061 | 0.035 | −0.170 | 0.085 | | |
| Gender | −0.736 | 0.520 | −0.086 | 0.158 | | |
| Nationality | 0.009 | 0.513 | 0.001 | 0.986 | | |
| Marital Status | 0.290 | 0.388 | 0.049 | 0.455 | | |
| Education level | −0.168 | 0.428 | −0.025 | 0.695 | | |
| Work department | −0.690 | 0.391 | −0.119 | 0.079 | | |
| Years of experience | 0.044 | 0.038 | 0.104 | 0.242 | | |
| Current position | 0.355 | 0.430 | 0.057 | 0.410 | | |
| Shift type | 0.383 | 0.439 | 0.066 | 0.384 | | |
| Hospital | −1.156 | 0.421 | −0.188 | **0.007**[*] | | |
| Motivation level | 0.005 | 0.009 | 0.033 | 0.604 | | |
| Psychological distress | 0.255 | 0.032 | 0.495 | **<0.001**[*] | | |

**Notes.**
  [*]p-value < 0.05.

**Table 6  Multiple regression analysis for variables associated with motivation level.**

| Variables | Unstandardized coefficients | | Standardized coefficients | Sig. | Model summary | |
|---|---|---|---|---|---|---|
| | B | Std. error | Beta | | $R^2$ | P value |
| (Constant) | 49.859 | 17.932 | | 0.006 | 0.103 | **0.046**[*] |
| Age | −0.366 | 0.279 | −0.149 | 0.192 | | |
| Gender | 4.024 | 4.139 | 0.068 | 0.332 | | |
| Nationality | 6.188 | 4.051 | 0.143 | 0.128 | | |
| Marital Status | 3.552 | 3.079 | 0.088 | 0.250 | | |
| Education level | −6.471 | 3.372 | −0.140 | 0.056 | | |
| Work department | 3.296 | 3.124 | 0.083 | 0.293 | | |
| Years of experience | 0.570 | 0.298 | 0.194 | 0.057 | | |
| Current position | 1.789 | 3.423 | 0.042 | 0.602 | | |
| Shift type | 3.185 | 3.489 | 0.080 | 0.362 | | |
| Hospital | −5.349 | 3.385 | −0.127 | 0.116 | | |
| Psychological distress | 0.234 | 0.296 | 0.066 | 0.430 | | |
| Work environment | 0.299 | 0.574 | 0.043 | 0.604 | | |

**Notes.**
  [*]p-value < 0.05.

## DISCUSSION

The main purpose of this study was to determine the relationships among psychological distress, work environment quality, and motivation among nurses working in Saudi Arabia. We revealed moderate levels of psychological distress, work environment quality, and motivation overall. These findings are similar to those of a previous study (*Davey et*
*al., 2019*) in which the authors revealed mild (12%) to moderate or severe (77%) levels of job-related stress among hospital nurses. The most frequent concerns were working in busy departments with increased duties and low salaries. These results also align with those of another study (*Simães & Rui Gomes, 2019*) wherein the authors reported that a high percentage of nurses have psychological distress (79.3%). Our findings are also consistent with those of a quantitative study wherein the authors revealed low to moderate levels of motivation (85.33 on a scale ranging from 19 to 133) (*Saleh, Eshah & Rayan, 2022*). Additionally, it has been found that motivation increases nurses' confidence and empowerment, while access to information enables nurses to make informed decisions about patient care, increasing their autonomy and leading to higher motivation levels.

Regarding the association between age and psychological distress, our findings are similar to those of previous studies, wherein the researchers conducted a cross-sectional study with nurses in Portugal (*Simães & Rui Gomes, 2019*). Their findings revealed that nurses with the oldest ages have more psychological distress. Our findings are different from another study's findings (*Wang et al., 2022*), which showed that there is no significant difference in psychological distress based on the age of participants. Therefore, psychological distress is not influenced by the age of nurses.

Psychological distress among registered nurses has been reported at 27.7% (*Belay et al., 2021*). These findings revealed predictive variables that put nurses at risk of developing psychological distress, including less working experience, poor communication with staff, fatigue due to the amount of work, and no social support. Nurses with extensive work experience also have more psychological distress than those with limited experience. It has also been found that nurse managers play a crucial role in establishing good relationships and communication among nursing staff, so building strong connections is essential for creating a positive work environment that can aid in reducing nurses' psychological distress (*Moore et al., 2013*).

Our findings indicate that work environment quality varies by department and hospital, which is consistent with the results of the study that *Patrician et al. (2022)* conducted. In their study, they compared work environment quality among the following hospitals: military, Magnet, Magnet-aspiring, and non-Magnet civilian hospitals. The authors found job satisfaction scores to be higher in military hospitals compared to other types of hospitals. They also showed that certain hospital types like Magnet hospitals may organically produce healthy work environments for nurses. Magnet hospitals can also provide opportunities for career advancement, professional development, and a supportive work culture. Based on the participants' responses, the work environment in private hospitals is more supportive of registered nurses' professional practice than the work environment in public hospitals (*Pires et al., 2018*). Work environment quality also varies among departments in terms of working hours, nurse–patient ratios, policies, the climate, leadership support, and the level of technology (*Mrayyan, 2009*).

For nurse managers, a higher educational level and an adequate monthly salary help boost motivation and job satisfaction, allowing nurse managers to handle various stressors at work more effectively (*Belay et al., 2021*). Moreover, our findings are similar to those of a systematic review in which the authors identified factors affecting

nurse motivation (*Baljoon, Banjar & Banakhar, 2018*). The review revealed that nurses' educational level affects their motivation at work. Most studies on this topic have also confirmed that health workers with a higher educational level (master's or doctorate) achieve maximum mean motivation scores. Further, factors that can contribute to low motivation levels among nurses include a lack of recognition or feedback, inadequate staffing levels, a lack of autonomy or control over their work, and high levels of stress and burnout. Such factors can lead to feelings of frustration, exhaustion, and a sense of purposelessness in their work (*Ramón et al., 2022*).

During the regression, nurses' nationality, current position, and work environment quality significantly influenced their psychological distress. Prior research has shown that foreign-born nurses experience higher psychological distress levels than their native counterparts (*Schilgen, Nienhaus & Mösko, 2020*). This might be attributed to factors like language barriers or cultural differences. There is also an association between nurses' current position and their psychological distress. That is, nurses in managerial positions may experience higher levels of psychological distress than staff nurses because the increased responsibilities and workload, coupled with the demands of managing a team, can contribute to higher levels of stress (*Niinihuhta et al., 2022*). Additionally, the relationship between the nursing work environment and psychological distress has been studied in a prior study. A significant association between poor work environments, which are characterized by factors like high workloads, a lack of control, and inadequate support, and higher levels of psychological distress among nurses has been reported (*Ren et al., 2022*). A negative work environment can also contribute to increased stress, burnout, and emotional exhaustion, which, in turn, can lead to psychological distress. High-quality hospitals like Magnet hospitals, as mentioned, have thus been recognized for their emphasis on nurse empowerment and better work environments compared to non-Magnet hospitals (*Lasater & Schlak, 2020*).

## Limitations of the study

Although our study provides valuable findings, it had a few limitations. First, a cross-sectional design can only be used to identify relationships among variables. Second, the use of self-administered questionnaires may have introduced response biases. Third, the generalizability of the findings may have been affected by our collecting data from only four institutions. Finally, using non-probability sampling (convenience sampling) might have led to selection bias. A replication of the study with a larger sample that represents more of the population can establish the findings' generalizability and external validity (*Matthay & Glymour, 2020*). Future studies should also be conducted on more public and private institutions, as well as in different cities. This would provide a better understanding of the disparities within the sample and serve as a foundation for improving the performance of nurses. Moreover, the findings of this study indicate potential areas that warrant further investigation related to the three studied variables and their potential predictors.

## Study implications

The study's findings suggest several strategies that nurse managers and leaders might employ in the work environment to increase motivation among nurses. These include establishing

a workplace that promotes nurses' involvement in decision-making, improves the work environment quality, expands internal opportunities, utilizes a reward and incentive system, and provides a greater work–life balance. Hence, to produce empowered and motivated nurses, healthcare organizations are advised to improve their work environment. Since nurses will function at their best when they have optimal mental health, administrators should attempt to assist nurses in maintaining psychological stability. Stakeholders should also take care of their nurses and teach them problem-solving skills and stress management strategies.

Further, developing strategies that limit the psychological distress among nurses can be helpful. Nurse managers may also support nurses by establishing fair work schedules, offering compassionate care, implementing group intervention tools like mindfulness classes, and promoting the use of healthy coping mechanisms to reduce levels of psychological distress and improve mental health. They can also prioritize reducing the psychological stress experienced by nurses since doing so will contribute to the improved quality of healthcare services and nurses' wellbeing.

## CONCLUSIONS

We concluded that nurses' work environment quality and motivation are interconnected with the psychological stress they experience at work. Monitoring nurses' work environment quality, motivation levels, and psychological distress is thus vital to ensure better patient care. Additionally, increasing work motivation will help to reduce nurses' psychological distress by increasing their work engagement, providing opportunities for professional development, and recognizing and rewarding excellence. Healthcare organizations should thus foster strategies to help motivate nurses in the workplace. Hence, future research on the long-term effects of motivation on staff's professional and personal lives is recommended.

### Funding

This work was supported by both the Researchers Supporting Project number (PNURSP2024R441) at Princess Nourah bint Abdulrahman University and the Researchers Supporting Project number (RSP2024R438) at King Saud University. The funders had no role in study design, data collection and analysis, decision to publish, or preparation of the manuscript.

### Grant Disclosures

The following grant information was disclosed by the authors:
Princess Nourah bint Abdulrahman University: PNURSP2024R441.
King Saud University: RSP2024R438.

### Competing Interests

The authors declare there are no competing interests.

## Author Contributions

- Hanan Alharbi conceived and designed the experiments, performed the experiments, authored or reviewed drafts of the article, and approved the final draft.
- Kholoud Alharbi conceived and designed the experiments, performed the experiments, prepared figures and/or tables, and approved the final draft.
- Ghareeb Bahari conceived and designed the experiments, analyzed the data, prepared figures and/or tables, and approved the final draft.
- Yousef Alshamlani analyzed the data, authored or reviewed drafts of the article, and approved the final draft.
- Regie Buenafe Tumala analyzed the data, prepared figures and/or tables, authored or reviewed drafts of the article, and approved the final draft.

## Human Ethics

The following information was supplied relating to ethical approvals (*i.e.*, approving body and any reference numbers):

Ethical approvals were obtained from the Institutional Review Boards of both Princess Nourah bint Abdulrahman University (Ref #: 22- 0445) and King Saud University (Ref #: 19/0737/IRB).

## Data Availability

The raw data is available in the Supplemental File.

## Supplemental Information

Supplemental information for this article can be found online at http://dx.doi.org/10.7717/peerj.18133#supplemental-information.

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
