# Peer review of "Psychological distress, work environment quality, and motivation levels among nurses working in Saudi Arabia"

_PeerJ, doi:10.7717/peerj.18133_

## Round 0.1 · original submission · Minor Revisions

Review the methodology and ensure consistency in how some terminologies are used. Also, it is recommended authors use the STROBE checklist to organize the intro and methods section. Finally, it might be of value for authors to have a native English speaker review the entire manuscript.

Reviewer 1 ·

Basic reporting

The article is clear, ethical, clear, explicit, and technically correct, and has sufficient introduction and background to demonstrate that the study fits into the knowledge domain of the research question.

Experimental design

The statistical method is clear. The research questions are well defined, relevant and meaningful. It illustrates how the research fills the identified knowledge gaps, and the submission clearly defines the research question, which is relevant and meaningful.

Validity of the findings

The results were valid and the underlying data were statistically reasonable and controllable. Conclusions should be properly stated, relevant to the original question being investigated and limited to questions supported by the results.

Additional comments

Approved for publication, minor changes required
Comments: A survey should be conducted on (foreign) nurses, mainly analyzing where they came from, the percentage of non-Saudis mentioned in the table, how they differ from each other, language proficiency, and whether it has any impact on the job. This needs to be subdivided and supplemented.

Reviewer 2 ·

Basic reporting

• This is an important paper regarding the field of psychology and work environment. Sufficient background/context were provided. However, provide number of nurses who were negatively affected by the psychological distress, and work environment.

Experimental design

The sample estimation was 142 from 4 hospitals. Please provide in detail how the sample was calculated.

Validity of the findings

• All underlying data have been provided; they are robust, statistically sound, & controlled.

Additional comments

Discussion: Line 262: “Furthermore, our findings reported that the work environment varies by department and hospital”. However, the multiple regression analysis for the department variable was not associated with work environment.

Reviewer 3 ·

Basic reporting

The current manuscript focuses on an important topic of psychological wellbeing of nurses in hospitals in Saudi Arabia.

1. One major issue of this manuscript is the grammatical errors. It is recommended that the authors review the article to correct the grammar errors and present information clearly throughout.
2. In the introduction, at lines 81-84, a citation (Sovold et al., 2021) was used to illustrate the negative impact of the COVID-19 pandemic on the well-being of nurses. However, the citation adopted is not focused on nurses. It is recommended to switch to an article that specifically focuses on the impact of the well-being of the nurses during the pandemic (example shown below).

Li, M., Yang, Y., Zhang, L., Xia, L., Zhang, S., Kaslow, N. J., Liu, T., Jiang, F., Tang, Y.L. & Liu, H. (2024). Mental health, job satisfaction and quality of life among psychiatric nurses in China during the COVID-19 pandemic: A cross-sectional study. Clinical Epidemiology and Global Health, 101540.

3. It is recommended to add hypotheses for each of the listed research questions in lines 106-111. Additionally, please discuss whether the results align or not align with the hypotheses in the Discussion section.

Experimental design

1. The descriptions of the Brief Nurses' Practice Environment scale is unclear, especially in lines 159-160 where the authors indicated the practice environment at "low levels" or "higher levels" based on the cutoff score. Please explain what the low and high levels mean in terms of the scores generated from this scale.

Validity of the findings

Another major issue of this manuscript is the presentation and interpretation of the results. For instance:
1. It is unclear what a "significant positive and moderate relationship" means in lines 218-219.
2. Lines 220-226 need revision. Please note the table(s) that illustrate these findings. In addition, please explain how the indicated variables are statistically different from the mentioned variables in lines 224-226.
3. Similarly, in lines 231-235, please add more information regarding the mentioned variables on how they significantly influenced psychological distress and work environment respectively.

---

## Round 0.2 · accepted · Accept

Based on peer reviewer's comments and my own assessment, I am pleased to inform you on the acceptance of your paper. Very soon, you will receive notification for copyediting. Congratulations!

Reviewer 3 ·

Basic reporting

The authors have addressed all the comments.

Experimental design

no comment

Validity of the findings

no comment